# A Research on Accident Reconstruction of Bus–Two-Wheeled Vehicle Based on Vehicle Damage and Human Head Injury

**DOI:** 10.3390/ijerph192214950

**Published:** 2022-11-13

**Authors:** Shang Gao, Mao Li, Qian Wang, Xianlong Jin, Xinyi Hou, Chuang Qin, Shuangzhi Fu

**Affiliations:** 1School of Petroleum Engineering, China University of Petroleum, Qingdao 266427, China; 2School of Chemistry and Chemical Engineering, Shanghai Jiao Tong University, Shanghai 200240, China; 3Aerospace System Engineering Shanghai, Shanghai 201108, China; 4School of Mechanical Engineering, Shanghai Jiao Tong University, Shanghai 200240, China; 5The Traffic Police Brigade of Shanghai, Municipal Public Security Bureau, Shanghai 200040, China

**Keywords:** bus–two-wheeled vehicle accident, reverse reconstruction, vehicle damage, head injury biomechanics, forensic identification

## Abstract

The problem of large calculation models in bus–two-wheeled vehicle traffic accidents (TA) leads to the difficulty of balancing the calculation efficiency and accuracy, as well as difficulties in accident reconstruction. Herein, two typical accidents were reconstructed, based on the rigid–flexible coupled human model (HM) and the Facet vehicle model, and the vehicle damage conditions and the human head biomechanical injury were analyzed. The simulation results showed that the physical process of the human–vehicle collision was basically consistent with the accident video, the windshield fracture was consistent with the actual vehicle report, and the human biomechanical injury characteristics were also consistent with the autopsy report, which verified the feasibility of the simulation model, and provides a basis and reference for forensic identification and for traffic police to deal with accident disputes.

## 1. Introduction

At present, two-wheeled vehicle accidents account for 28% of world traffic fatalities and make up 20% of all traffic deaths in China [1]. In recent years, study of the perception and protective equipment of two-wheeled vehicles under complex traffic conditions has also received increasing attention from researchers [2,3,4]. Qin [5] made a statistical analysis of 297 two-wheeled vehicle accidents recorded by Shanghai Traffic Police Corps, and the statistical data are shown in Appendix A. It was found that bus–two-wheeled vehicle TAs account for a large proportion of the total types of accidents (30%), and second only to car–two-wheeled (cars in this paper refer to passenger cars) vehicle TAs (50%). The lower limbs and head of the cyclist are the most vulnerable body parts, accounting for 27% and 24%. However, bus accident reconstruction research has mainly focused on bus–pedestrian collisions [6,7,8,9,10], bus–car collisions [11,12], and collision of multiple buses [13,14]. Two-wheeled vehicle accident reconstructions mainly focus on car–two-wheeled vehicle collisions [1,15]. Reconstruction of TAs involving a bus–two-wheeled vehicle accident is rarely performed. It is necessary to reconstruct and analyze bus–two-wheeled vehicle accidents and focus on the fatal head injury in a simulation study.

The overall mass of a bus is large and its outer contour surface is rigid and approximately a vertical plane, resulting in different cyclist rebound directions and human injuries compared with car–two-wheeled vehicle collisions. When a two-wheeled vehicle is added into an accident reconstruction, this makes the reconstruction of bus–two-wheeled vehicle accident dynamic trajectory and the research on human injuries more complicated and difficult. Guo et al. [16], based on the multi–body (MB) method, reconstructed the collision process of a bus–two-wheeled vehicle accident and analyzed the head HIC value of the cyclist. However, this method cannot analyze the vehicle damage condition and the distribution of injury von Mises stress (VMS), and also cannot provide a reference for the study of biomechanical injury mechanisms [17,18].

The types of TAs are diverse, and different reconstruction methods are also required for TA research. The most commonly used methods include MB modeling, which can describe the trajectory of the vehicle during the collision; finite element (FE) modeling, which can analyze the mechanism of human injury and the condition of vehicle damage; and the method of co–simulation [19,20,21,22,23,24,25,26]. However, the existing accident reconstruction methods have limitations [19]. The MB method cannot accurately simulate biomechanical injuries such as abrasions and fractures. The calculation efficiency of the FE method is low. Co–simulation involves different software types and the input and transformation between parameters is complex, which may make the final human injury analysis inaccurate. Therefore, it is particularly important to construct an accident reconstruction model that takes into account both impact kinematics (efficiency) and human biomechanical injury mechanisms (accuracy).

In view of the above requirements, this paper first establishes a new Facet bus model. This model can calculate vehicle deformations and save computational time. Second, a rigid–flexible coupled HM that can analyze human FE injury and improve computing efficiency is constructed. Finally, this human–vehicle–road coupling model is used in accident reconstruction, to analyze the physical process of a human–vehicle collision, the vehicle damage condition, and the human biomechanical injuries. The above analysis results were compared with the accident video, vehicle scene photos, and forensic injury reports. The results showed that the motion posture of the cyclist and bus after the collision were consistent with the accident situation, and the biomechanical injury of the human body and the forensic appraisal report were mutually supporting, which can provide a basis for traffic police to deal with accident disputes.

## 2. Materials and Methods

We used MADYMO 7.5 to reconstruct two TAs. We chose the Facet model to build the vehicle and a rigid–flexible coupled model to build the HM. Finally, data and pictures in MADYMO were extracted and compared with the actual injury.

### 2.1. Facet Vehicle Model

Herein, the vehicle model used was the Facet model, which is a FE mesh of empty materials. The Facet model is a three–node or four–node unit and is completely attached to a reference space, rigid body, or deformed body. It not only ensures accuracy but also saves calculation time. The Facet model was built by importing and exporting to/from 3Dmax, HYPERMESH, and MADYMO, in turn, and setting the relevant grids and nodes [1,27,28]. For a Facet model of accident vehicles, such as buses, in the contact coupling calculation, it is necessary to define the equivalent elastic–plastic stiffness curve of the vehicle collision; that is, the relationship between the contact force and penetration, as shown in Figure 1 [29,30].

In MADYMO software (version 7.5, Siemens, Berlin, Germany), the Facet model can realize the connection and relative motion between different parts. Different parts such as the windshield and wheels can be split to make the simulation process closer to the real situation. The model customization process is shown in Figure 2.

### 2.2. Rigid–Flexible Coupled HM

The rigid–flexible coupled HM is based on the TNO international standard MB HM and the HUMOS international standard FE HM. It is split according to the head, trunk, upper limbs, and lower limbs, and then combined using a MB and a FE local human body model. Figure 3 is the rigid–flexible coupled modular HM mixed structure customization, the split MB part model and the FE part model are connected using the support method through joint and constraint.

### 2.3. Head FE Model

In this study of bus–two-wheeled vehicle accidents, a FE model of the head was used. The human head and FE head structures are shown in Figure 4, which is a three–dimensional model composed of human tissues, such as bones and skin. The head material parameters used in this study are shown in Table 1. Through the local blunt and accident simulation experiments, it was verified that the contact of each body part of the rigid–flexible coupled HM can function normally and that the contact characteristics of the model conform to the contact characteristics of the human body, which proved the validity of the model [1].

## 3. Case Reconstruction and Results 

The cases selected were a side collision, which accounts for the highest proportion of two–wheeled vehicle TAs, and a scratch collision, which accounts for the second highest proportion [5]. The two TA cases were simulated, to reconstruct the physical process of human–vehicle collision, analyze the vehicle damage condition and human biomechanical injury, and compare the simulation results with CCTV footage, to verify the validity of the model, so as to ensure the accuracy of vehicle speed in the simulation model.

### 3.1. Case 1: Electric–Bicycle (E–Bike)–Bus Scratch Collision

#### 3.1.1. Accident Information

This TA occurred on a narrow road. The scratch collision occurred due to the bus occupying the road and due to the blind spot of the bus driver. As shown in Figure 5, the bus and the E–bike were driving in the same direction. The E–bike was in front and the bus came from behind. Then the front door of the bus scraped the E–bike cyclist, causing the cyclist to fall onto the side of the bus. The bus rolled the cyclist under the bus chassis as it continued to travel, causing the cyclist’s head to fall to the ground, and then the rear wheel of the bus crushed the cyclist’s head, resulting in the death of the cyclist on the spot. The focus of this accident simulation was the speed of the bus and E–bike at the moment of collision. The speed at the time of the collision was considered to be the basis for determining whether the vehicle was speeding. 

According to the accident file data of the Shanghai Traffic Police Brigade, we could determine the brand and model of the bus and E–bike. The bus was an ordinary large passenger car and the E–bike was a portable electric bicycle. The information of the accident vehicles is shown in Table 2. The cyclist in the TA was a 61–year–old female, who was 160 cm in height and weighed 60 kg. According to forensic medical identification, the cyclist’s head was crushed by the bus wheels, resulting in cranial bone fractured, which is a fatal injury, leading to the death of the cyclist.

#### 3.1.2. Determination of Initial Accident Parameters

To determine the initial relative position and initial collision speed of the bus and E–bike before the scratch accident, the photogrammetric method was used. The principle of photogrammetry is to select a certain reference datum in the photograph, and at the same time, using a known length as a reference, project the key points and key lines into a two–dimensional coordinate system; then you can obtain the length and position coordinates of each line segment. Two typical video screenshots were selected for measurement, as shown in Figure 6 and Figure 7.

During the measurement process, the wheelbase of the front and rear wheels of the bus was used as the measurement benchmark, the design standard for the wheelbase of this model of bus was 6 m, and the distance traveled by the bus between time 1 and time 2 was 3.779 m, and the time interval between time 1 and time 2 was 0.48 s; thus, the initial speed of the bus could be calculated to be about 7.87 m/s. Similarly, it can be calculated that the speed of the E–bike was about 5.96 m/s. The MB and FE coupling contact algorithm in MADYMO software was used to set the contact characteristics between people, vehicles, and roads. The friction coefficient between the wheels of the bus and the ground was defined as 0.7, and the friction coefficient between the wheels of the E–bike and the ground was also set as 0.7. The friction coefficient between the cyclist and the ground was 0.6, the friction coefficient between the cyclist and E–bike was 0.5, and the friction coefficient between the cyclist and bus was 0.3. Then, taking the cyclist’s motion posture and the position of the landing point as the objective function [18], the simulation results were obtained through multiple iterations.

#### 3.1.3. Accident Reverse Reconstruction

In this TA, the head injury was fatal, so a female head FE combined HM was adopted. This model combined the advantages of (i) saving calculation time and improving calculation efficiency, and (ii) analyzing the FE VMS distribution of the brain and improving the simulation accuracy. Both the E–bike and the bus used the Facet vehicle model, and the simulation model of Female HM and Facet vehicle is shown in Figure 8.

The motion postures of the cyclist and vehicle after scratch collision were obtained in the simulation, as shown in Figure 9. The front door of the bus first made contact with the left arm of the cyclist; then the cyclist was tripped and fell from the E–bike into the bus and collided with the ground; at this time, the bus continued to drive, and finally the rear wheel of the bus rolled over the head and other parts of the cyclist. The simulation animation results were basically the same as in the video surveillance, including the rest position of the cyclist and the E–bike, which were basically consistent with the actual situation. This further proved the accuracy of the human–vehicle–road coupled model and the feasibility of the rigid–flexible coupled HM in reconstructing the ergonomic response. After reconstruction of the TA, it could be determined that both the bus and E–bike were speeding. The main cause of this TA was the blind spot for bus drivers.

#### 3.1.4. Result Analysis

A simulated cranial bone VMS cloud diagram is shown in Figure 10. The simulation results showed that the maximum VMS of the cranial bone was 8.63 × 10^7^ Pa. Based on the VMS criterion of Tian et al. [31], when the stress on the human head exceeds 7.5 × 10^7^ Pa, a fracture will occur and cause death. The simulation results far exceeded the stress tolerance limit of the human head, indicating that the cranial bone of the cyclist would have been fragmented, and the injury cloud map can accurately show the location of the largest stress, so as to judge that the position of the cranial bone fragment was located in the occipital bone of the posterior brain. The autopsy result was that the cyclist’s cranial bone was severely fragmented, and the head injury was fatal. The simulated head injury results of the cyclist were consistent with the autopsy results.

Figure 11 shows the head acceleration curve of the cyclist. It can be seen that when t = 70 ms, the head acceleration reaches the maximum value, and the head injury is the greatest. The head injury criteria (HIC) value was 3667, far exceeding the standard value of human safety 1000 [32]. The injury data were consistent with the FE analysis and the corpse report, further verifying the reliability of the rigid–flexible coupled HM in the human FE injury analysis.

### 3.2. Case 2: Bicycle–Bus Side Collision

#### 3.2.1. Accident Information

One afternoon in 2009, the weather was clear and the road surface was dry. There was a bus driving from the south (the bottom of the figure) to the north (the top of the figure) at a crossroad in a city. A construction worker rode a bicycle at the intersection and turned left. When the bus driver spotted the bicycle, he performed an emergency brake, but because the bus was too fast, it did not brake in time and hit the bicycle. After the bicycle riding east collided with the front left of the bus, the cyclist and the bicycle were ejected forward and rolled over onto the ground. The bus slid forward about 3 m and then stopped. The scene of the accident is shown in Figure 12. The focus of this TA simulation was the speed and position of the bus at the moment of collision. The collision speed was considered as the basis to decide whether the driver was speeding. Whether the bus was on the pedestrian zebra crossing at the moment of collision is an important basis for the determination of accident liability.

According to the accident file, the left side bumper of the bus had no obvious scratches and dents. There were radioactive cracks on the lower left side of the front windshield. The right pedal of the bicycle was broken, and the right handlebar and tires were deformed. The cyclist was a 39–year–old male who was 165 cm in height and a medium build. The accident resulted in the fracture of the cyclist’s right cranial bone, multiple abrasions from the left face to the neck, dislocation of the right ankle, and multiple abrasions all over the body. The vehicles involved in the TA were an ordinary bus and bicycle. The basic information of the vehicles is shown in Table 3.

#### 3.2.2. Determination of Initial Accident Parameters

In this TA, the vehicle had no obvious brake marks, so it was impossible to infer information such as the speed of the vehicle before the collision based on brake marks. However, a traffic surveillance camera took video data of the accident at the intersection, and the photogrammetric method can also be used to determine important information, such as vehicle speed and orientation, which provided a reference for the accident simulation. The method referred to case 1 and performed a three–dimensional reconstruction of the accident scene, as shown in Figure 13. The standard zebra crossing was used as the calibration object, and the braking of the bus was regarded as a uniform deceleration process. The speed of the bus at the time of collision was about 15.8 m/s; the speed exceeded 50 km/h, which would threaten the live of a cyclist.

#### 3.2.3. Accident Reverse Reconstruction

The model and parameter settings refer to case 1. A 50th percentile male rigid–flexible coupled model was selected as the HM, and it was scaled according to the size of the victim. The ordinary bicycle model was selected as the two–wheeled vehicle model. According to the information provided by the monitoring video record and the three–dimensional reconstruction of the accident scene in PHOTOMODELER software (2021 version Canada, Vancouver), the vehicle collision orientation and collision speed were fine–tuned, and the results were obtained after repeated iterative calculation, as shown in Figure 14. After the bus collided with the bicycle, the cyclist and the bicycle fell to the ground, the cyclist was ejected forward, to stop about 2.7 m away from the bus, and the bus slid forward about 2.3 m and then stopped. The simulation results are basically consistent with the actual rest position of the cyclist and bicycle. According to the reconstruction of the accident and the results of the investigation, it could be determined that the bus had exceeded the pedestrian zebra crossing when it started to brake. Therefore, the cause for the accident was the speeding of the bus.

#### 3.2.4. Result Analysis

The simulation results in Figure 14 show that, after the cyclist’s right shoulder collided with the bus, his head swung out to the right and hit the windshield, the head was impacted for the first time, and then he fell to the ground, impacting the head for the second time. Figure 15 is the head acceleration curve of the cyclist during the collision. It can be seen that at t = 180 ms, when the head hits the windshield, the head acceleration reaches 4500 m/s^2^, and the head injury is the greatest. The HIC of the cyclist’s head was 3976, exceeding the human safety standard value of 1000 [32]. When the cyclist collided with the ground, the head acceleration reached 1500 m/s^2^, causing a secondary injury to the human body. The head injury process was consistent with the dynamic response process. The injury data obtained from the simulation were consistent with the injury report of the cyclist’s concussion.

A VMS nephogram of the cranial bone is shown in Figure 16. It can be seen that the maximum VMS of the right skull was 9.45 × 10^7^ Pa, which is far beyond the stress tolerance limit, indicating that the cranial bone of cyclist would have been fragmented. In the injury nephogram, it can be accurately judged that the location of cranial bone fracture is in the right brain, which is consistent with the right temporal contusion. The autopsy found that a head injury was the fatal injury to the victim. An injury comparison of the two cases is shown in Table 4. The simulated head injury results of the cyclists in the two cases were consistent with the autopsy results.

As shown in the screenshot of the simulation animation in Figure 17a, when t = 160 ms, the cyclist’s right calf hit the bumper of the bus. Figure 17b is a torque diagram of the tibia. The data shows that the maximum torque of the calf at the time of collision reached 180 N·m, and the maximum shear force reached 2.9 kN, both of which were less than the injury limit of the human body [33]. This is consistent with the report that the cyclist’s ankle had a dislocation but no fracture.

There are only a few kinds of FE material in MADYMO software, so in the Facet bus model, the front windshield was separated, meshed in HyperMesh, and the relevant properties were set according to Wang et al. [34]. Figure 18a shows a diagram of the broken position and shape of the bus windshield, which is consistent with the crack position on the left bottom edge of the windshield in Figure 18b. Therefore, the accuracy of accident reconstruction was further proven.

## 4. Conclusions

In conclusion, this study provides an insight into the reconstruction of bus–two-wheeled vehicle TAs. It solved the problems in bus–two-wheeled vehicle TAs, whereby (i) bus modeling is complex and accident reconstruction is difficult; (ii) computational efficiency and accuracy are difficult to balance; and (iii) the proportion of data is relatively large, and that in the research is relatively less. Compared with previous FE or MB models, this model can take into account both the efficiency and accuracy. This method also provides a new concept for the dynamic simulation and biomechanical injury mechanism analysis of bus–two-wheeled vehicle accidents.

Although the collision kinematics and human injury of bus–two-wheeled vehicle accidents were reconstructed through accident case studies, there are still deficiencies in this study. Due to the limitation of material types in MADYMO, the cracking of glass could not be simulated. Herein, only a broken shape morphology distribution was used to replace this. Another deficiency was the lack of big data analysis of bus–two-wheeled vehicle TAs. The next step should be the analysis of multiple TA cases, summarizing the characteristics of the bus–two-wheeled vehicle accident collision dynamics and head injuries, and comparing them with car–two-wheeled vehicle accidents, to identify differences in the accident characteristics. Specific suggestions could put forward for the improvement of the structure and material of buses, as well as human protection devices (such as helmets, etc.), preferably with the design of an effective collision–avoidance system. Moreover, the human biomechanical injuries could be transformed into medical language, to provide effective application data.

## Figures and Tables

**Figure 1 ijerph-19-14950-f001:**
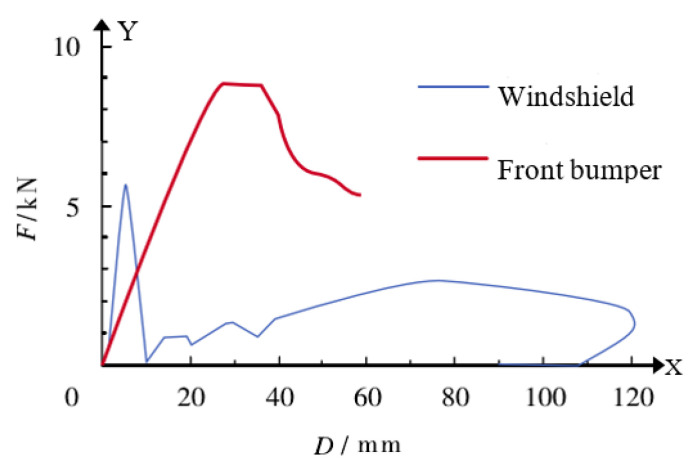
The equivalent stiffness curve of the vehicle surface.

**Figure 2 ijerph-19-14950-f002:**
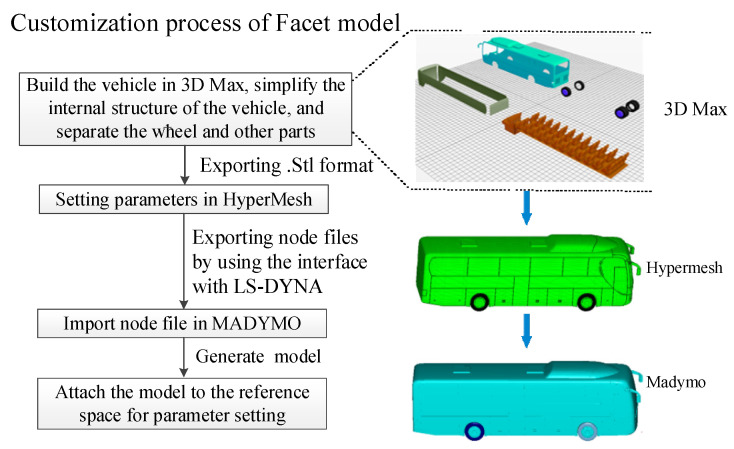
The model customization process.

**Figure 3 ijerph-19-14950-f003:**
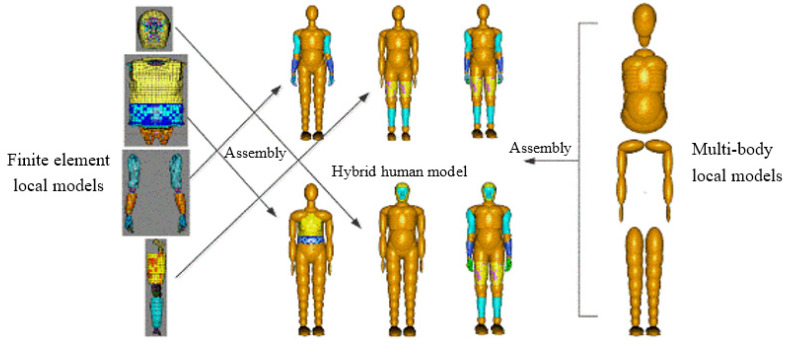
Rigid–flexible coupled modular HM mixed structure customization.

**Figure 4 ijerph-19-14950-f004:**
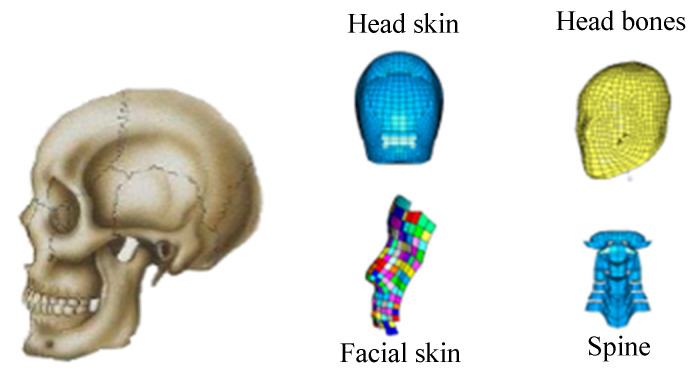
Diagram of human head (**left**) and FE head structure (**right**).

**Figure 5 ijerph-19-14950-f005:**
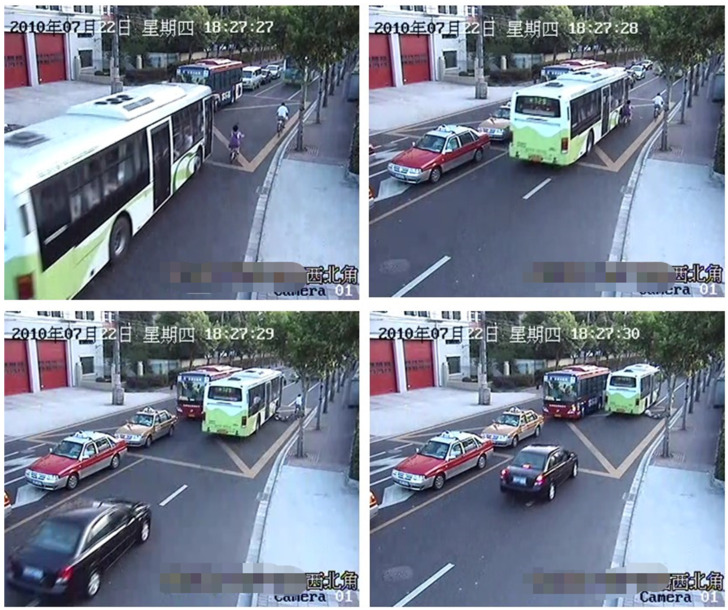
The scene of accident case 1 (the time shown in the picture is Thursday, 22 July 2010).

**Figure 6 ijerph-19-14950-f006:**
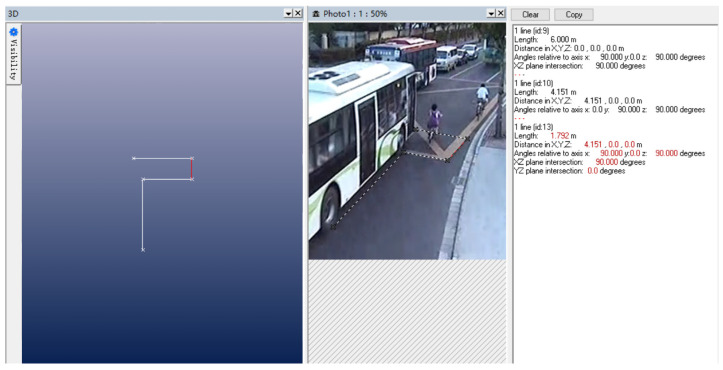
Initial collision time 1.

**Figure 7 ijerph-19-14950-f007:**
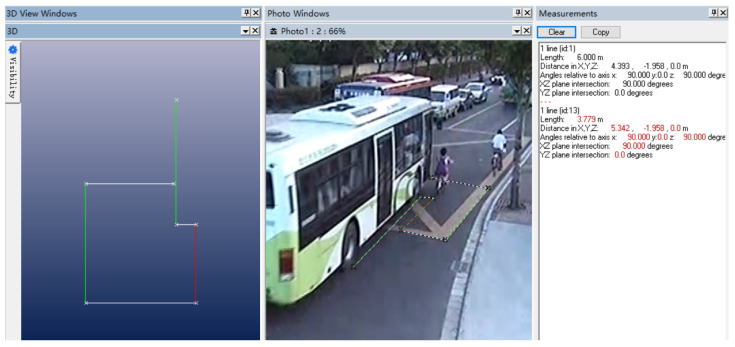
Initial collison time 2.

**Figure 8 ijerph-19-14950-f008:**
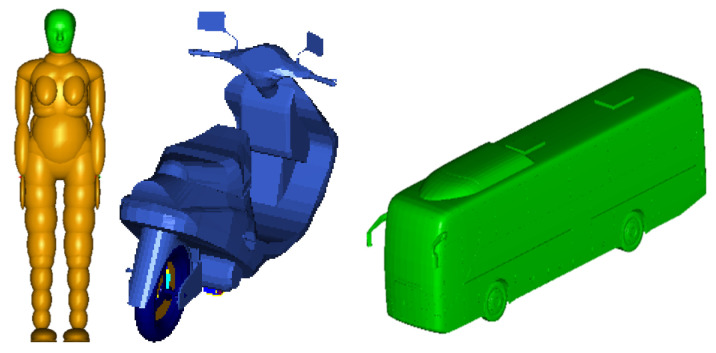
The simulation model of Female HM (**left**) and Facet vehicle (**right**).

**Figure 9 ijerph-19-14950-f009:**
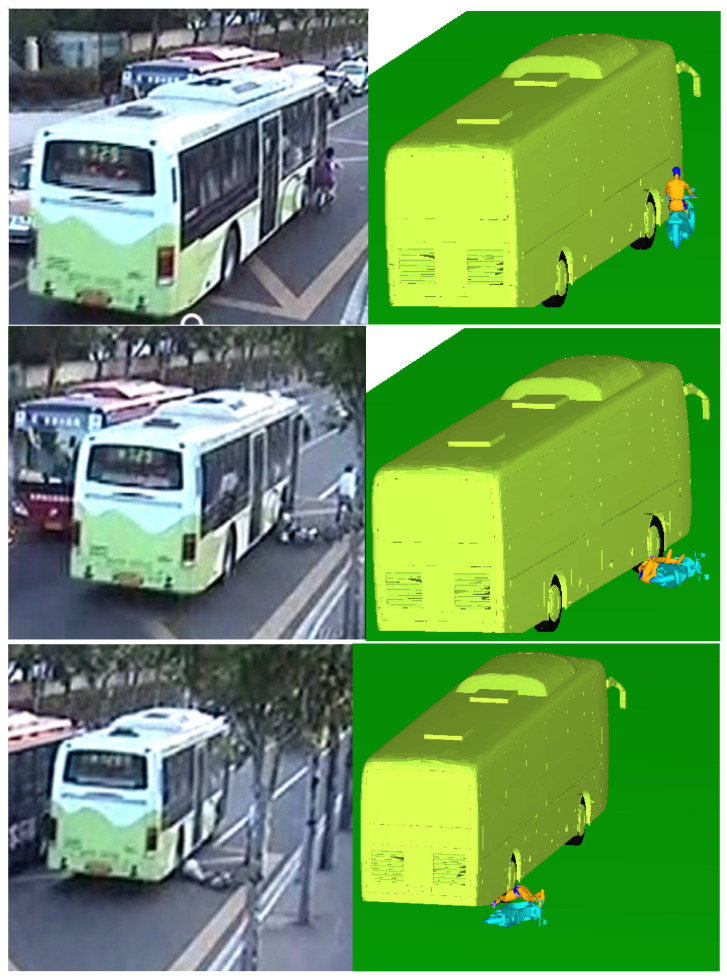
Comparison of the accident simulation results with the real conditions in case1.

**Figure 10 ijerph-19-14950-f010:**
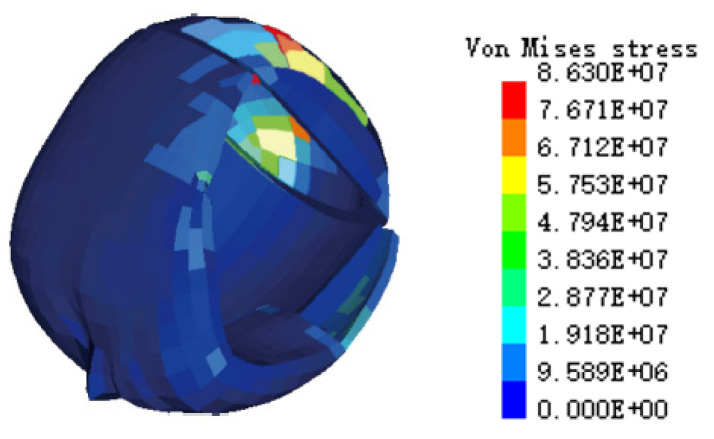
The head injuries of the cyclist cyclone VMS in case 1.

**Figure 11 ijerph-19-14950-f011:**
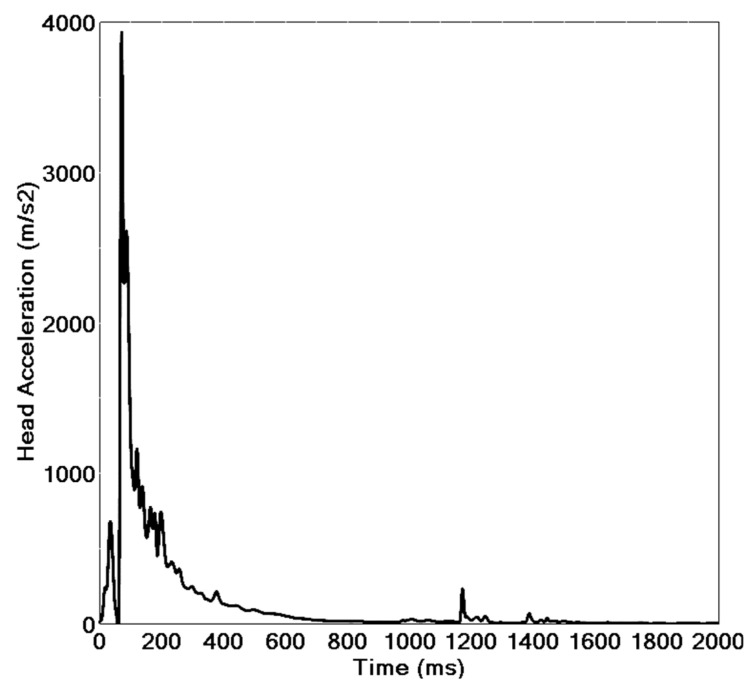
Acceleration curve of the cyclist’s head.

**Figure 12 ijerph-19-14950-f012:**
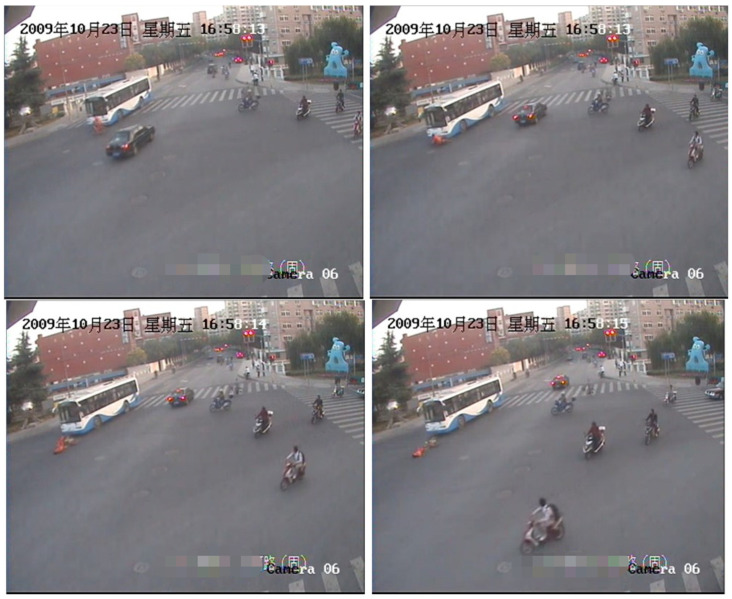
Scene of the accident in case 2 (the time shown in the picture is Friday, 23 October 2009).

**Figure 13 ijerph-19-14950-f013:**
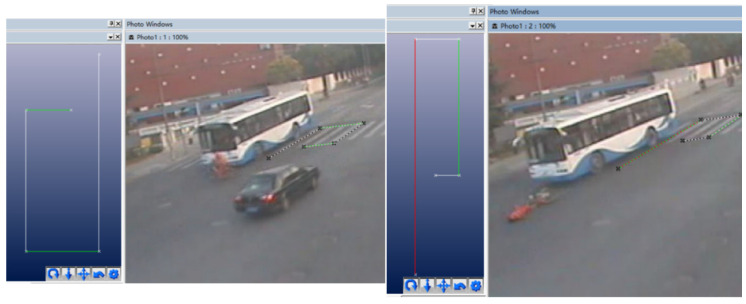
The position of the bus before (**left**) and after the collision accident (**right**).

**Figure 14 ijerph-19-14950-f014:**
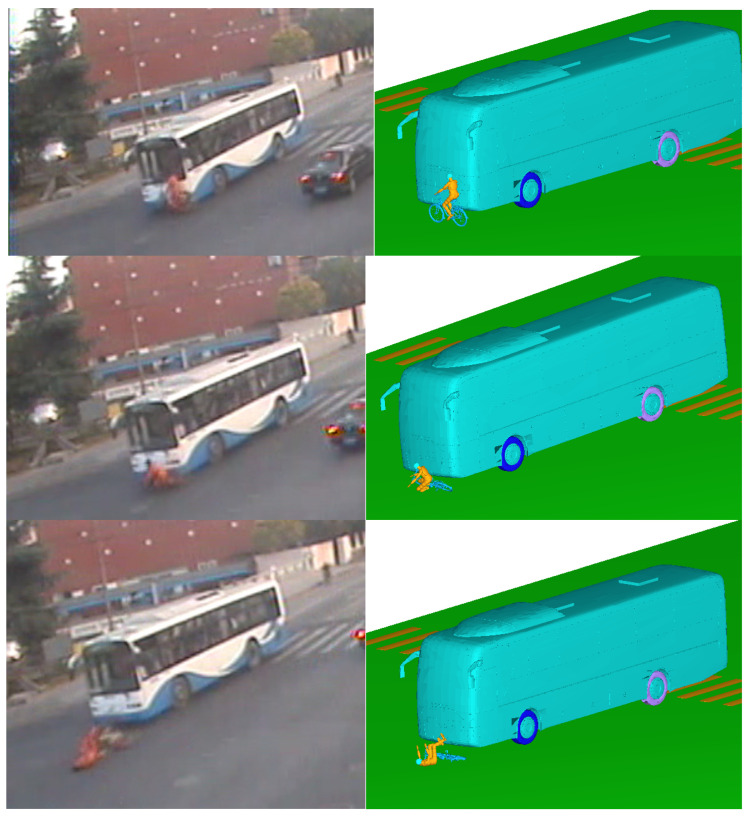
Comparison of the accident simulation results with the real conditions in case2.

**Figure 15 ijerph-19-14950-f015:**
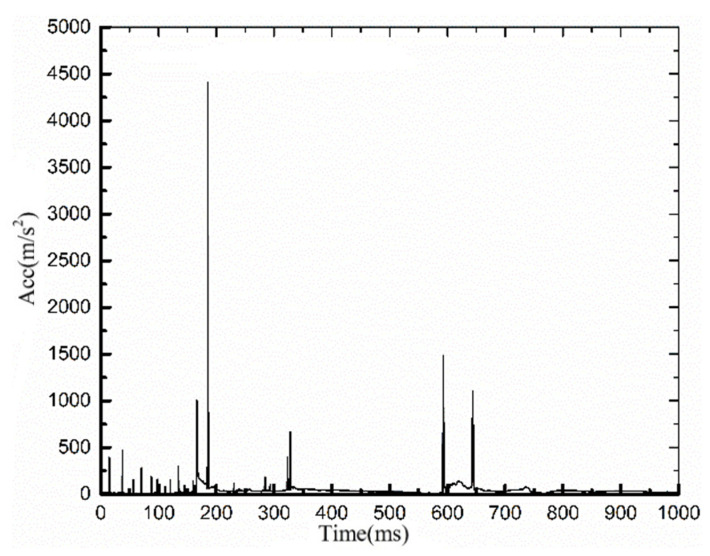
The resultant acceleration curve of the cyclist’s head.

**Figure 16 ijerph-19-14950-f016:**
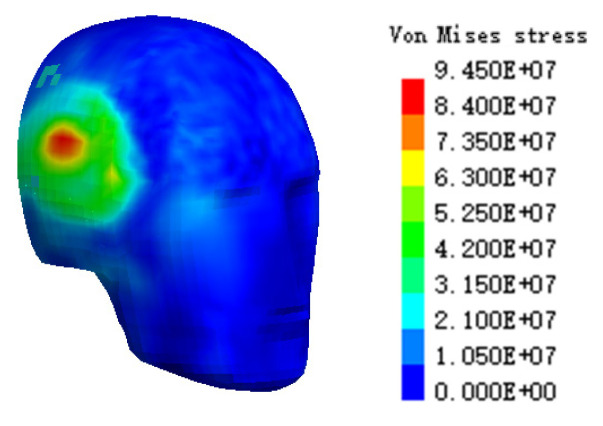
The head injuries of cyclist cyclone VMS in case 2.

**Figure 17 ijerph-19-14950-f017:**
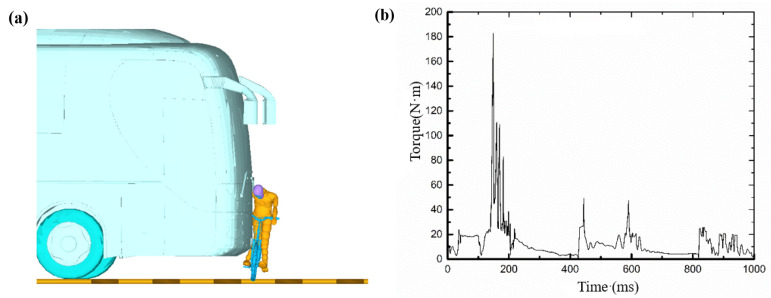
Injury analysis of right lower limb: (**a**) the moment of lower limb collision, (**b**) torque curve of the tibia.

**Figure 18 ijerph-19-14950-f018:**
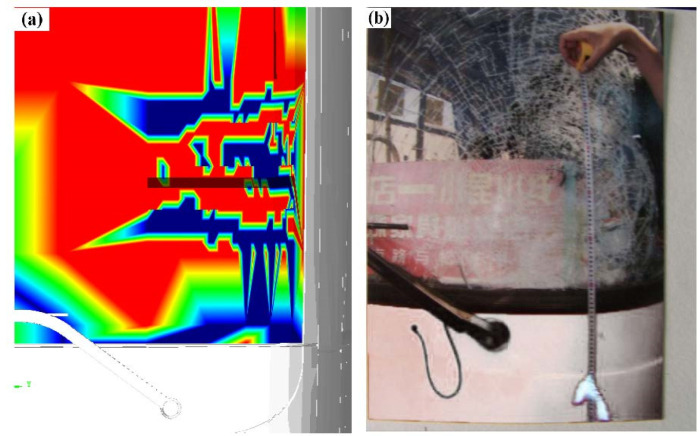
Diagram of the broken bus windshield: (**a**) simulated crack morphology distribution, (**b**) actual crack morphology distribution.

**Table 1 ijerph-19-14950-t001:** Head material parameters used in this study.

Component	Parameters				
	Material Model	Density (kg·m^−3^)	Youngs Modulus (MPa)	Poisson’s Ratio	Yield Stress (MPa)
Spine	elastic	1133	31.5	0.45	-
	Material model	Density (kg·m^−3^)	Shear modulus (MPa)	Poisson’s ratio	Yield stress (MPa)
Head skin	ogden_rubber	1135	0.13	0.499	-
Facial skin	ogden_rubber	1135	0.04	0.499	-
	Material model	Density (kg·m^−3^)	Shear modulus (MPa)	Bulk modulus (MPa)	Yield stress (MPa)
Skull	elastic_plastic	1110	420	537.6	32.7

**Table 2 ijerph-19-14950-t002:** The physical appearance parameters of the vehicles.

Vehicle Type	Length/Width/Height (mm)	Wheelbase (mm)	Quality (kg)
Bus	10,000/2550/3250	6000	11,800
E–bike	1180/–/510	–	60

**Table 3 ijerph-19-14950-t003:** Physical appearance parameters of the vehicles.

Vehicle Type	Length/Width/Height (mm)	Wheelbase (mm)	Quality (kg)
Bus	12,000/2550/3250	6000	11,800
Bicycle	1680/–/510	–	18

**Table 4 ijerph-19-14950-t004:** Comparison of the injury results.

Evaluation Criteria	Parameter	Tolerance Limit Value	Simulation Value	Autopsy Results
Evaluation criteria based on MB	HIC	1000	3667 (case1) 3976 (case2)	Occipital bone fracture
Evaluation criteria based on FE	VMS of cranial bone (Pa)	7.5 × 10^7^	8.63 × 10^7^ (case1) 9.45 × 10^7^ (case2)	Right skull fracture with concussion

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
