# Peer review of "A Research on Accident Reconstruction of Bus–Two-Wheeled Vehicle Based on Vehicle Damage and Human Head Injury"

_ijerph, 2022, doi:10.3390/ijerph192214950_

Round 1

Reviewer 1 Report

The authors reconstructed two typical accidents based on the rigid–flexible coupled human model and the Facet vehicle model, and the vehicle damage condition and the biomechanical injury of the human head are analyzed and validated with the two practical accidents. I think this work is interesting and can be published in this journal. However, the authors should make major revisions to address the listed issues before it is accepted.

1.       All the material presented is more like a technical report than a scientific manuscript, especially for Chapter 2 (materials and methods). Also, the title of this chapter does not match the material presented in this section. Please rename and reorganize this section by detailing and presenting both finite element models, material properties, and boundary conditions for both models used in the FE analysis.

2.       One of the goals of this topic is accident reconstruction of bus-two-wheeled vehicles based on vehicle damage and human head injury that is not well explained. It is not clear how to reconstruct this type of accident if there is no video footage of the accident. If the accident is captured by a video camera, there is no point in performing numerical simulations.

3.       Another goal of this paper is to present a method of mixed finite element model for solving such kind problems by using numerical simulations, which is also not well explained.

4.       In Fig. 1 information from experimental research carried out by other authors is presented but does not show how these data were used in the finite element models of this investigations.

5.       Many figures are not informative; for example, see Fig.3, Fig.4, Fig.8 and Fig.10. Lef sides of the pictures in Fig. 6, Fig.7, and Fig.13 are also not informative.

6.       Several figures with not correct titles, for example, Fig.2, Fig.4, Fig.8 and Fig.18. Therefore, there is also no legend in Fig. 18. (a).

7.       The information provided in Tables 1 and Table 2 is not informative.

8.       In lines 209 and 210 it states 'The value of the head injury criteria (HIC) is 3667, far exceeding the standard value of human safety 1000'. There is no reference to the value of HIC 1000.

9.       In lines 297 - 299 it states that 'As shown in the screenshot of the simulation animation in Fig.17(a), when t=160ms, the right calf of the cyclist hits the bumper of the bus. Fig.17(b) is the torque diagram of the cyclist“, but the title of this figure states „The analysis of the injury of the right lower extremity (a) the moment of collision of the lower extremity, (b) the torque curve of the tibia“. Which statement is true?

10.   In lines 299 - 301 it states: 'The data show that the maximum torque of the calf at the time of collision reached 180N·m, and the maximum shear force reaches 2.9kN, both of which were less than the injury limit of the human body.' Provide limited values and references to these values.

11.   Conclusions are not quite correct and informative. It is more like a summary of the manuscript than the conclusions of this work.

Reviewer 2 Report

The authors reconstructed the two fatal bus-two-wheeled vehicle collision by both multibody and FE models. As the authors described, accident reconstruction is the important for considering both impact kinematics and human biomechanical injury mechanisms. Because the results were well accordance with the autopsy results according to the values of HIC or VMS, these results confirmed the validity of this methos. Obtained results might be valuable for establishing the preventive measures for fatalities.

However, in this study, although the authors proposed the procedures for reconstructing bus-two-wheeled vehicle collision, each method such as using MADYMO or FE human models was popular as previously reported. Furthermore, we can easily understand the kinematics or mechanisms of collision according to the vehicle and human injuries or the videos without simulations. In these two cases, as the applied forces were too large, it is difficult to save the victims after collisions. If the authors propose an effective collision- avoidance system with the simulation, the manuscript would achieve the level of publication of this Journal.

The reviewer also has following ethical concerns: the name of the road or the collision spot were shown in the Fig. 5 and 12; the face of the victims were shown in Fig. 10 and 16 without retouch.

Reviewer 3 Report

The paper includes examples of a very complex and interesting topic. The paper is beautifully written, but full of technical errors. I make suggestions for improving the paper:

-        Not all abbreviations used in the paper are defined (etc. FE).

-        The introduction is readable and includes interesting references. I'd like you to describe the importance of two-wheeler perception equipment in the introduction and cite a couple of recent studies:

Damani, J., & Vedagiri, P. (2023). Following Behavior of Motorized Two-Wheelers in Mixed Traffic Conditions. In Conference of Transportation Research Group of India (pp. 63-82). Springer, Singapore.

Simović, S., Ivanišević, T., Trifunović, A., Čičević, S., & Taranović, D. (2021). What affects the e-bicycle speed perception in the era of eco-sustainable mobility: a driving simulator study. Sustainability, 13(9), 5252.

Mohammadi, E., Azadnajafabad, S., Keykhaei, M., Shakiba, A., Meimand, S. E., Shabanan, S. H., ... & Rahimi-Movaghar, V. (2022). Barriers and factors associated with the use of helmets by Motorcyclists: A scoping review. Accident Analysis & Prevention, 171, 106667.

 -        In Figure 1, the names of the x and y axes are missing (only units are shown).

-        Please indicate whether the listed software is licensed.

-        The results show examples of two traffic accidents. I have a big concern, about whether the above data, including the figures, can be used for public publication. I am also concerned, about whether explicit photos and parts of footage of traffic accidents are allowed to be shown in the journal. If there is an ethical/legal framework for publication, please state in one sentence that you have adhered to ethical norms.

      Use "passenger car" instead of "car".

-        The conclusion is very meager. Please give specific recommendations, on how the results of your study can be used in practice (e.g. for traffic accident expertise). Are these simulations credible in a court of law in your country? It would be interesting, in future research, to compare the results of simulating a traffic accident with the results of a manual calculation of an analysis of a traffic accident.

Round 2

Reviewer 1 Report

The authors have added Section 2.3 “Head FE model” and added the material properties of the following table, however, this section does not present the finite element of the head model as is common in finite element analysis. Although the table shows the properties of the materials, for head and facial skin Youngs modulus is not presented and Poisson’s ratio is indicated as 0.5 which is impossible in finite element analysis.
